Facilitative and competitive interaction components among New England salt marsh plants

Bruno John F. jbruno@unc.edu 1
Rand Tatyana A. 2
Emery Nancy C. 3
Bertness Mark D. 4
1 Department of Biology, University of North Carolina at Chapel Hill , Chapel Hill , NC , United States of America
2 Northern Plains Agricultural Research Laboratory, USDA-ARS , Sidney , MT , United States of America
3 Department of Ecology and Evolutionary Biology, University of Colorado at Boulder , Boulder , CO , United States of America
4 Department of Ecology and Evolutionary Biology, Brown University , Providence , RI , United States of America
Toonen Robert
Electronic publication date: 2017 Nov 29
Publication date: 2017
Volume: 5
Electronic Location ID: e4049
Received 2017 Aug 4; Accepted 2017 Oct 26
Copyright: ©2017 Bruno et al.
Copyright year: 2017
Copyright holder: Bruno et al.
License: This is an open access article distributed under the terms of the Creative Commons Attribution License, which permits unrestricted use, distribution, reproduction and adaptation in any medium and for any purpose provided that it is properly attributed. For attribution, the original author(s), title, publication source (PeerJ) and either DOI or URL of the article must be cited.
License URL: https://creativecommons.org/licenses/by/4.0/

Keywords: Competition, Facilitation, Interaction strength, Salt marsh

Funding: National Oceanic and Atmospheric Administration Office of Ocean Coastal Resource Management graduate research fellowships National Science Foundation This research was funded by National Oceanic and Atmospheric Administration Office of Ocean and Coastal Resource Management graduate research fellowships to John F. Bruno and Tatyana A. Rand and a National Science Foundation grant to Mark D. Bertness. The funders had no role in study design, data collection and analysis, decision to publish, or preparation of the manuscript. There was no additional external funding received for this study.

==============================
Intra- and interspecific interactions can be broken down into facilitative and competitive components. The net interaction between two organisms is simply the sum of these counteracting elements. Disentangling the positive and negative components of species interactions is a critical step in advancing our understanding of how the interaction between organisms shift along physical and biotic gradients. We performed a manipulative field experiment to quantify the positive and negative components of the interactions between a perennial forb, Aster tenuifolius, and three dominant, matrix-forming grasses and rushes in a New England salt marsh. Specifically, we asked whether positive and negative interaction components: (1) are unique or redundant across three matrix-forming species (two grasses; Distichlis spicata and Spartina patens, and one rush; Juncus gerardi), and (2) change across Aster life stages (seedling, juvenile, and adult). For adult Aster the strength of the facilitative component of the matrix-forb interaction was stronger than the competitive component for two of the three matrix species, leading to net positive interactions. There was no statistically significant variation among matrix species in their net or component effects. We found little difference in the effects of J. gerardi on Aster at later life-history stages; interaction component strengths did not differ between juveniles and adults. However, mortality of seedlings in neighbor removal plots was 100%, indicating a particularly strong and critical facilitative effect of matrix species on this forb during the earliest life stages. Overall, our results indicate that matrix forming grasses and rushes have important, yet largely redundant, positive net effects on Aster performance across its life cycle. Studies that untangle various components of interactions and their contingencies are critical to both expanding our basic understanding of community organization, and predicting how natural communities and their component parts will respond to environmental change.

Introduction

Species interactions are often composed of both negative (competitive) and positive (facilitative) components (Callaway, 1994; Greenlee & Callaway, 1996; Callaway & Walker, 1997; Claus Holzapfel & Mahall, 1999; Stachowicz, 2001; Crotty & Bertness, 2015). The relative importance of these components is likely to shift along gradients of physical stress and consumer pressure (Altieri, Silliman & Bertness, 2007; Altieri & Van de Koppel, 2014; Crotty & Bertness, 2015). For example, competition between common acorn barnacles for space in rocky intertidal habitats (Connell, 1961) is a classic illustration of the role played by competition in determining fine-scale distribution patterns within natural communities. But at high tidal heights (Bertness, 1989; Bertness et al., 1999a) and in warmer climates (Lively & Raimondi, 1987; Bertness et al., 1999b; Leonard, 2000), barnacle neighbors actually facilitate one another by buffering against desiccation stress. Examples from desert (Muller, 1953; Niering, Whitaker & Lowe, 1963), chaparral (Callaway, Nadkarni & Mahall, 1991), and salt marshes (Bertness & Hacker, 1994; Callaway, 1994) illustrate that plant interactions also commonly shift from competitive to facilitative across physical stress gradients.

Our current knowledge of the balance between the positive and negative components of species interactions is limited. One specific issue that needs further exploration is how this balance changes ontogenetically across the life history stages of the interacting organisms to determine the net effect of each species on the other. In many plants (Muller, 1953; Niering, Whitaker & Lowe, 1963; Callaway, 1994; Bruno & Kennedy, 2000; Rand, 2000; Yelenik, DiManno & D’Antonio, 2015) and sessile invertebrates (Dayton, 1975; Bertness & Grosholz, 1985; Leonard, 1999), recruits and juveniles depend on neighbors for early survival and growth, but as adults they may primarily compete with these same neighbors (Niering, Whitaker & Lowe, 1963; Bertness & Grosholz, 1985; Bertness & Yeh, 1994; Callaway, 1995). In this situation, common in stressful environments, facilitation of juveniles may establish clumped adult distribution patterns that are dominated by competitive forces. In other cases, the situation is reversed: for example, a number of studies have found stronger competitive effects at early life history stages (emergence or recruitment), with interactions becoming neutral to facilitative at later life history stages (adult survival) (Thomson, 2005; Leger & Espeland, 2010; Rojas-Sandoval & Meléndez-Ackerman, 2012). Additionally, whether interactions are predominately competitive or facilitative at a given life stage can depend strongly on the environmental context (Rand et al., 2015), e.g., the intensity of physiological stress and degree of resource limitation.

More generally, we still know very little about how positive and negative interaction component strengths vary among species. Is species identity an important consideration or are species redundant in the positive and negative effects they exert on their neighbors and the community as a whole? The answer to this question is likely to depend on the specific mechanisms by which competition and/or facilitation operate among species, which is in turn likely to vary depending on the study system examined. For example, the effect of nurse shrubs on seedling survival and growth in Mediterranean environments have been shown to depend strongly on shrub identity, e.g., due to morphology (Kraft et al., 2014), suggesting a lack of functional redundancy among facilitators (Gómez-Aparico et al., 2004). In contrast, two sea grass species were found to have broadly similar (i.e., redundant) effects on the composition of associated fouling communities (Moore & Duffy, 2016).

In salt marsh plant communities, positive interactions are strongly driven by neighbors shading one another from physical stress (Bertness & Yeh, 1994; Angelini et al., 2015), in which case morphologically similar species might be predicted to have similar positive effects on neighbors. However, the competitive components of these neighbor interactions will depend on the resources being competed for and the morphological and physiological adaptations of the competing species. Thus, in salt marshes, the positive components of species interactions may be redundant, while the competitive components may not, leading to a de-coupling of the components.

To date, experiments separating interaction components have only been performed in a few systems (Greenlee & Callaway, 1996; Claus Holzapfel & Mahall, 1999), and to our knowledge no studies have examined how interaction components shift ontogenetically across the life history of the species or with the identity of organisms in the same functional group. The purpose of this study was to answer two questions related to the context-dependency of species interaction components: (1) do the strength of the positive and negative components of the interaction between a perennial salt marsh forb, Aster tenuifolius, and matrix forming grasses and rushes, change with the identity of the matrix species (Juncus gerardi, Distichlis spicata, and Spartina patens)? and (2) does the strength of these components vary with life stage of A. tenuifolius (seedling, juvenile, and adult)?

Methods

Study system

We conducted a field experiment at Nag Creek marsh on Prudence Island, Rhode Island, USA (41°37′41.0″N 71°19′04.4″W), to examine the components of interactions among plant species in a southern New England salt marsh system. Nag Creek is typical of New England salt marshes, which are characterized by dense stands of perennial grasses and rushes that form bands, or zones, across the tidal gradient (Niering & Warren, 1980; Nixon, 1982; Bertness & Ellison, 1987). Variation in the frequency of flooding results in a strong gradient in salinity and soil oxygen availability (redox potential) corresponding to tidal height (Bertness & Hacker, 1994; Hacker & Bertness, 1999). The seaward marsh zone is dominated by the grass Spartina alterniflora, which is replaced by Spartina patens and then a rush, Juncus gerardi, with increasing elevation and distance from the shoreline. Another grass, Distichlis spicata, is patchily distributed within the S. patens and J. gerardi zones (Miller & Egler, 1950; Bertness & Ellison, 1987). This species is generally abundant only in areas of high disturbance or increased physiological stress, where competition with the zonal dominants is absent (Brewer & Bertness, 1996).

A group of less abundant halophytic forbs (herbaceous dicots) are generally found interspersed within the matrix of dominant grasses and rushes. Many of these forbs experience strong competitive suppression by the dominant matrix species which can limit both seedling recruitment and adult plant survival and reproduction (Ellison, 1987; Shumway & Bertness, 1992; Brewer, Levine & Bertness, 1998; Rand, 2000). Under stressful conditions, however, these same grasses and rushes ameliorate soil conditions and have a net facilitative effect on forbs (Bertness & Shumway, 1993; Bertness & Hacker, 1994). Facilitation results primarily from shading of the substrate by the vegetation canopy which reduces surface evaporation and the accumulation of salt on the soil surface (Bertness, Gough & Shumway, 1992; Callaway, 1994), but possibly also by oxygenation of the soil (Bertness, 1991). The degree to which different matrix marsh species vary in their competitive or facilitative effects on salt marsh forbs is not known, in part because earlier work has been done within natural zone communities and did not differentiate between effects of neighbor identity and tidal elevation (Hacker & Bertness, 1999; Rand, 2000). In addition, while previous studies have examined the net effects of matrix species on forbs, the relative strengths of the competitive and facilitative components of the grass-forb interaction have not been quantified in this system.

Our experimental target species was Aster tenuifolius (Asteraceae), a relatively salt-tolerant perennial that germinates and emerges in early spring, flowers in late fall, and produces small wind-dispersed seeds following reproduction. This species is abundant in both the S. patens and J. gerardi zones in New England salt marshes and is also frequently found in association with D. spicata. (Brewer, Levine & Bertness, 1998; Rand, 2000). We manipulated the species identity and structure of the matrix vegetation surrounding seedling, juvenile, and adult Aster plants to test if the facilitative, competitive, and net interaction effects on Aster are affected by neighboring plant identity, neighboring plant density, or the life stage of Aster individuals. We were specifically interested in the unidirectional effects of matrix species on marsh forbs, such as Aster, and not the reciprocal effects.

Experimental design

Our experimental design included three treatments, with individual replicates of each treatment clustered spatially into blocks within each matrix species: (1) an unmanipulated control, (2) a neighbor removal, and (3) a facilitation mimic that duplicates the positive effects of the neighbor without any of the negative effects (Fig. 1). In control treatments, vegetation was left intact. In the vegetation removal treatment all above ground vegetation within each 0.25 m2 plot was clipped at the substrate surface with scissors once every two weeks until regrowth no longer occurred. In the facilitation mimic treatment, all above ground vegetation was once again removed to eliminate competition for light. Plots were then covered with 0.25 m2 pieces of shade-cloth that were pinned to the substrate surface using plastic staples. Shade-cloth reduces soil surface evaporation to effectively mimic the facilitating effects of plant neighbors (Bertness, Gough & Shumway, 1992). Shade-cloth was spray-painted flat white to avoid soil temperature increases that often occur under black cloth in the field. Light measurements above the removal and facilitation mimic plots (Fig. 2) indicate light conditions did not differ between these treatments and were not meaningfully affected by the manipulation, e.g., due to greater reflectance of the white shade.

Figure 1 Experimental design.

Photograph of a treatment block using J. gerardi as the transplanted matrix species with four experimental treatment plots (see labels) and a single Aster adult transplanted into each treatment plot. Photo credit: John Bruno.

Figure 2 Environmental conditions and starting plant biomass.

Effects of the matrix species (A, C, E) and experimental manipulations (B, D, F) on environmental conditions (data are means ±1 SE from July measurements; August data are not presented but were nearly identical) and initial Aster biomass.

We manipulated the identity of the matrix species, while controlling for environmental context, by transplanting square plots (0.25 m2 in area, 50 cm deep) of the three matrix species (S. patens, J. gerardi, and D. spicata) into the J. gerardi (high marsh) zone in the Nag Creek marsh during the winter of 1999. Transplanting all matrix species into a single marsh zone allowed us to standardize tidal height so that any differences between species could be attributed directly to neighbor identity. Thirty Spartina patens and D. spicata plots were removed and placed into equal-sized holes in the J. gerardi zone. J. gerardi plots were lifted out of the soil and then replaced to simulate transplantation. Transplanted plots were grouped into ten blocks with three plots/block for the D. spicata and S. patens matrix transplants and the J. gerardi plots used for Aster seedlings and juvenile plants (10 blocks for each). For the J. gerardi - adult Aster plots (20 blocks), we included an additional fourth treatment in each block: a thinning treatment in which vegetation was thinned to 50% of its original density by clipping with scissors.

In mid-May of 1999, after the transplanted matrix plots had established in their new locations and experimental treatments were in place, Aster individuals were collected from natural populations at Nag Creek and a single target adult plant (>2 true leaves) was transplanted into the center of each plot of each matrix species. Seedlings were identified by the presence of cotyledons, and juvenile plants were one-year-old, pre-reproductive individuals with <2 true leaves and no cotyledons present. The mean (±1 SE) height (cm) and dry weight (mg) of the three experimental life stages (n = 100) were as follows: seedling height = 0.72 ± 0.07, seedling weight = 1.64 ± 0.12; juvenile height = 6.49 ± 0.21, juvenile weight = 24.97 ± 1.58; adult height = 16.61 ± 0.44, adult weight = 146.45 ± 5.27. In the vegetation mimic treatments, Aster transplants were inserted through a slit that was made in the center of the shade-cloth, and the cloth was then pinned back together around the stem using plastic staples. All Aster transplants were watered for the first week after planting to minimize transplant shock, and plants that died within the first two weeks after transplantation were replaced. The experiment was set up by June 1, 1999 and was harvested in August 1999. Peak Aster biomass at the study site is in late August/early September (Brewer, Levine & Bertness, 1997). Thus we maximized the experimental growing season, yet harvested plants before they began to drop leaves and lose biomass.

Salinity and light levels were quantified in each plot on July 20 and August 20, 1999, to evaluate the effects of the experimental treatments on potential salt stressors and above-ground competition. Salinity measurements were taken by extracting a core of peat, 3 cm diameter × 3 cm deep, in each matrix plot. We pressed each sample through cotton gauze cloth and quantified salinity of the extracted pore water using a hand-held NaCl refractometer (precision = ±1 g kg−1). Light levels were measured between 10:00 AM and 2:00 PM with a LiCor solar monitor (Model 1776). Instantaneous measurements (µE m−1) were taken 5 cm above the soil surface (the height of the sensor) and above the canopy in each experimental plot. To estimate the initial mass of Aster transplants, we measured the longest leaf length of each experimental Aster on June 17, 1999. Longest leaf length was also measured on an additional 82 juvenile and 43 adult Aster individuals that were then harvested and weighed the same month. A regression equation between longest leaf length and biomass was used to estimate initial biomass of experimental plants (adults: y = 0.04039 × –0.13501, adjusted R2 = 0.77; juveniles: y = 0.01433x − 0.02160, adjusted R2 = 0.76). Final biomass was estimated by harvesting all experimental plants at the end of August, drying them to a constant mass into a drying oven at 55 °C for one week, and weighing each plant to the nearest milligram. Relative growth was calculated for each plant as: (measured final biomass − predicted initial biomass)/predicted initial biomass.

We used three simple equations to calculate the positive and negative components within each block: strength of positive component=mimic−removal

strength of negative component=control−mimic

strength of net component=control−removal.

Interaction strength components were calculated using both final mass and relative growth. We compared the absolute value of each interaction component among matrix species and Aster life stages with one-factor ANOVA (block was not included as a treatment since each block produces a single value for each interaction type).

Figure 3 Final Aster biomass.

Final biomass of Aster plants after 75 days in the experimental treatments and different matrix species. The thinned treatment, in which grass shoot density was reduced by 50%, was only applied in the plots with Aster adults transplanted into the J. geradi matrix. Data are means ±1 SE. Final Aster biomass varied significantly (P < 0.05) among treatments for all three matrix species (One-factor ANOVA).

Results

There was little variance in light reduction across the three matrix species (Fig. 2). The experimental treatments manipulated light and salinity levels as intended. The matrix vegetation reduced light at the soil surface to just less than 20% ambient, and thinning J. gerardi roughly doubled the amount of available light to ∼40% ambient (Fig. 2). Vegetation removal resulted in a substantial increase in salinity levels from a mean of 46.8 g kg−1 (±1.3, SE) in mimic and vegetation treatments to 71.3 g kg−1 (±2.2) in vegetation removal treatments (Fig. 2). Thinning of the J. gerardi canopy had no significant effect on salinity and the full canopy of all three matrix species reduced salinity to similar levels (Fig. 2). Vegetation mimics were effective in increasing light transmittance to levels similar to those in removal treatments while maintaining salinity at levels similar to those within natural vegetation (Fig. 2), suggesting this treatment effectively replicated the salinity reduction aspect of facilitative interaction component.

Due to mortality of some of the experimental Aster transplants, our sample size was reduced to nine for juvenile Aster (in J. gerardi), eight for adult Aster in S. patens, and 14 for adult Aster in J. gerardi. None of the Aster transplants in D. spicata were lost. Mortality of Aster seedlings in the matrix removal treatments was 100%, 0% in the vegetation (“Intact”) plots, and 58% in the facilitation mimic plots, hence seedling data could not be used in the primary analysis, i.e., we could not calculate interaction component strengths for the seedling stage. Initial adult Aster biomass, estimated from regression equations (described above), did not differ between the three matrix species, or between neighbor manipulation treatments (Fig. 2). By the end of the experiment growth varied significantly between treatments (Fig. 3). Regardless of the matrix species, the strength of the positive component of the matrix-forb interaction outweighed the negative component, resulting in a positive net effect of the matrix species on Aster adults and juveniles (Fig. 4). There was no significant variation among matrix species in their net effects or interaction components, however, the negative effect of J. gerardi appeared negligible compared to that of D. spicata (Fig. 4). Calculation of interaction component strengths based on final mass and relative growth (Fig. S1) were qualitatively similar. Within the J. gerardi matrix, relative growth of adult and juvenile Aster was lower in removal treatments than in mimic, thinned, and control (intact vegetation) treatments, and did not differ significantly among the later three treatments (Fig. 3). The effects J. geradi neighbors did not differ between Aster juveniles and adults.

Figure 4 Species interaction components.

Net, negative, and positive effects of three salt marsh matrix species on adult and juvenile Aster based on final biomass. Apparent differences in the strength of interaction components among the matrix species were not statistically significant (P > 0.05, ANOVA).

Discussion

Our results indicate that the net effect of matrix species on forbs consists of both positive and negative components, the strength of which did not differ significantly among matrix species or between later Aster life stages. The strength of the positive component was generally stronger than the negative component (except for D. spicata) resulting in a positive net effect of the matrix species on Aster. Previous studies have defined J. gerardi to be the “keystone facilitator” in this system (Hacker & Gaines, 1997). The strength of the facilitative component was as strong for D. spicata and S. patens as it was for J. gerardi, suggesting that other matrix species can play a similar functional role as J. gerardi.

Thinning J. gerardi blades by 50% significantly increased light availability to Aster transplants but had no effect on salinity levels or Aster growth rates. Adult Aster growth in the thinning plots was similar to growth in intact vegetation. The relatively weak (but not significantly different) competitive effect of J. gerardi may in part be explained by the fact that it grows and flowers earlier than the two grass species, and then dies back relatively early in the growing season. This could reduce competition with forbs later in the summer. Moreover, the blades of J. gerardi tend to decompose rapidly, thus it does not form the dense thatch of standing dead biomass, that is characteristic of both S. patens and D. spicata canopies. These traits could reduce the strength of the positive and negative effects of J. gerardi on its neighbors. In any case, the fact that S. patens, J. gerardi, and D. spicata reduce salinity to similar levels and positively affect Aster growth to a similar degree suggests that all three species serve a strong facilitative function in New England salt marshes. Thus, in this system the role of matrix species as facilitators appears to be largely redundant. This is consistent with the hypothesis that species from the same functional group have similar effects on other functional types within the same community; however, we expect that other functional groups, such as forbs or shrubs, are likely to have different net effects (and relative contributions of competitive and facilitative components).

The interpretation of our experiment assumes that all grass and rush matrix species share a common mechanism of facilitation of marsh forbs—the reduction of soil salinity by shading the substrate. However, some marsh matrix species also aerate the soil and hide neighbors from herbivores (Ellison, 1987; Hacker & Bertness, 1999), which could serve as other mechanisms for enhanced forb fitness. The same is true in many cases of plant-plant competition because neighboring individuals are often simultaneously competing for multiple resources (e.g., light, water, nutrients). Including more than one facilitation or competition mimic treatment to simulate the effects of other facilitative or competitive mechanisms could be used to tease part the strength and context-specificity of each individual mechanism. On the other hand, if the main interest is to determine the cumulative effect of all facilitative or competitive mechanisms, the interaction mimics will have to be designed to include all known mechanisms.

We found little difference in the effects of matrix species on the performance of Aster at different life history stages. For example, there were no differences between either raw growth rates or interaction component strengths between juvenile and adult Aster. Mortality of seedlings in J. gerardi removal plots was 100%, and thus interaction component strengths could not be calculated for this life stage. However, the high mortality clearly indicates that there is a strong and critical facilitative effect of matrix species on this forb during the earliest life stages, a finding concordant with other empirical studies (e.g., Kennedy & Bruno, 2000). One of the largest drawbacks of our experiment was the high within-treatment variance that reduced our power to detect differences among treatments. Statistical power could be improved by increasing the sample size, however, under conditions where mortality of individuals may be exceedingly high, as found for the seedling life stage in the present study, we suggest that within-treatment variability could be reduced by setting up replicate response individuals (here, Aster transplants) in each plot. For individual-level response variables like growth or fecundity, a plot-wide mean could be calculated. This approach would also allow for the calculation of population-level parameters such as percent survivorship and the inclusion of species especially susceptible to environmental stress (because even total mortality in the neighbor removal treatment would produce a continuous variable which is necessary for the calculation of component strengths).

Conclusion

Despite the difficulties involved, studies designed to tease apart the importance of positive and negative components of interactions and their contingencies represent an exciting venue of research with the potential to greatly expand our understanding of community organization. Understanding such issues is not just an academic exercise—it is essential in order to predict how natural communities and their component parts respond to environmental heterogeneity. Furthermore, we cannot predict how the structure and organization of natural communities will respond to climate change and anthropogenic stresses until we understand how different components of the species interactions respond to such changes (Bertness et al., 1999a).

Supplemental Information

Figure S1 Species interaction components based on relative growth

Net, negative, and positive effects of three salt marsh matrix species on adult and juvenile Aster based on relative growth. Apparent differences in the strength of interaction components among the matrix species were not statistically significant (P > 0.05, ANOVA).

Click here for additional data file.

Additional Information and Declarations

Competing Interests

Author Contributions

Data Availability

John F. Bruno is an Academic Editor for PeerJ.

John F. Bruno conceived and designed the experiments, performed the experiments, analyzed the data, wrote the paper, reviewed drafts of the paper.

Tatyana A. Rand conceived and designed the experiments, performed the experiments, wrote the paper, reviewed drafts of the paper.

Nancy C. Emery conceived and designed the experiments, performed the experiments, wrote the paper, prepared figures and/or tables, reviewed drafts of the paper.

Mark D. Bertness conceived and designed the experiments, performed the experiments, reviewed drafts of the paper.

The following information was supplied regarding data availability:

Bruno, John (2017): Data for “Facilitative and competitive interaction components among New England salt marsh plants” (Bruno et al 2017, PeerJ). figshare.

https://doi.org/10.6084/m9.figshare.5263093.v1.

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
