# Peer review of "Facilitative and competitive interaction components among New England salt marsh plants"

_PeerJ, doi:10.7717/peerj.4049_

## Round 0.1 · original submission · Minor Revisions

I am sorry for the delay in getting this back to you, we have had more trouble than usual in trying to find referees. Overall both referees are enthusiastic about the work, while also having a number of suggestions for improvement. The more significant of the criticisms is that ecological investigations into competition and facilitation have progressed significantly since 1999, and this paper should include more up-to-date references and better explain how this research fits within current ecological understanding. The referee leans toward this being a major revision, whereas the first referee has nearly the same evaluation, but leans toward that being a minor revision. I find myself firmly in the middle, but given that the requested revision does not require reanalyses or reinterpretation of the data, I believe that can be accomplished with a good edit, and so I am returning a decision of minor revision.
I believe that each of the referees has some good points and useful suggestions for improvement, both leaning towards a more general and contemporary framing of the results for those outside the salt marsh system, that I expect will improve the manuscript. I look forward to seeing your revised manuscript.

·

Basic reporting

This paper describes a well-designed experiment that tests the degree to which plant-plant interactions are positive and negative across the life history of a perennial forb and determines the redundancy of three matrix species. The analyses are appropriate. The writing is clear and professional. The authors cite the relevant literature both within and outside their study system to support their approach.

To prepare the reader for the methods, the authors should be more specific in the introduction regarding the meaning of light environment and salinity. They should also be more detailed about the importance of (reasoning behind) the 'mimic' treatment.

Specific comments for the introduction:
Line 81 an e.g. or example is needed here

Line 89 examples of species specific factors that preclude functional redundancy should be included here.

Experimental design

no comment

Validity of the findings

The authors are careful in interpreting their results, but so much so that there is little that those working outside the salt marsh system can take away from the paper. The authors should add context to their conclusions: how do these results apply to other ecological systems? Improving the text supporting the light and the salinity measurements will help with this.

Additional comments

none

Reviewer 2 ·

Basic reporting

ABSTRACT:
Lines 27, 29: “Three matrix-forming grass and rush species” implies three of each. Use instead, “one species of rush and two grass species” for clarity.

Line 31: Suggest changing “forbs” to simply the species name Aster tenuifolius since this was the only forb analyzed in the experiment, and “forbs” implies more than one forb species.

Line 33-35: “There was no statistically significant variation among matrix species
in their net or component effects, however, the competitive effect of J. gerardi was
negligible, especially compared to that of D. spicata.” Unclear wording.

Line 34: “…net or component effects. However, …”


INTRO:

Line 50: Although elucidating facilitation within your study system is the main aim of the research, this first line should be inclusive of both general types of interaction (facilitation and competition) as the concept is introduced initially. I.e. “Research on community organization in ecology has long recognized that species interactions are often composed…”. Then, honing that down through this first paragraph to the last lines,

Lines 63-65: Try “The prevalence of facilitation in instances of environmental stress or particular life-history stages does not minimize the role played by competition…” Omit “trivialize”, I’m not sure any ecologist would ever consider competition a ‘trivial’ component of a community, but I think I get at your point.

Line 99: ”…may not, leading to a de-coupling…”

Lines 107-108: Same comment as above in abstract regarding clarity of grasses and a single rush species. Also, comma unnecessary after “rushes”


METHODS
Line 113-114: It might be good to include the nearest town or other geographical landmark (or Lat/Long?) to specifically identify Nag Creek Marsh. Being unfamiliar with the area, it took a bit of time to locate the marsh online.

Line 143: You state that A. tenuifolius both is a perennial and that it germinates in early spring. An individual of A. tenuifolius from a previous season is not ‘germinating’, but rather re-emerging from established below-ground rootstock. Is there a difference in emergence/germination times?


DISCUSSION
Line 265: “…standing dead biomass, …”

Experimental design

Line162-163: “Shade cloth…in the field.” The white painting may have provided elevated photosynthesis (over either darker bare substrate or surrounding vegetative matrix). How was this either accounted for or monitored?

Lines 170-171: Were the dug J. gerardii plots replaced in the exact same hole from which they were dug, or were they moved to one of the four plots within the block to simulate hole size or local substrate mismatch likely part of transplanting the other matrix species?

Lines 188-189: is this growth timeline (June to August) the full breadth of seasonal growth of these species? There needs to be more life-history information within the Introduction, or here, to show why these times were chosen. The Introduction identifies “early spring” as the germination time (and see my previous comment regarding perennials), and August seems early for the end of growth (thinking more September/October).

Validity of the findings

No Comment.

Additional comments

It is noted that this research was done in the summer of 1999. The majority of the references for this work are 15 or more years old, with a few exceptions. Before publication, I strongly recommend including more current research on the topics included, with attendant descriptions of how this work remains novel and contributory within that context.

---

## Round 0.2 · accepted · Accept

Having read through the revision and your response to referees, I believe that you have addressed their concerns, and see no reason that it should not be moved forward into production.